

# A 50-year analysis of hydrological trends and processes in a Mediterranean catchment

Nathalie Folton, Eric Martin, Patrick Arnaud, Pierre L'Hermite, Mathieu Tolsa

Irstea, RECOVER Research Unit, 3275 Route de Cézanne, CS 40061, 13182 Aix-en-Provence, France.

*Correspondence to*: Nathalie Folton (nathalie.folton@irstea.fr)

**Abstract.**

The Réal Collobrier hydrological observatory in southeastern France, managed by Irstea since 1966, constitutes a benchmark site for regional hydro-climatology. Because of the dense network of stream gauges and raingauges available, this site provides a unique opportunity to evaluate long term hydro-meteorological Mediterranean

trends. The main catchment (70 km²) and its sub-catchments are located in the Maures massif of Southeastern France, close to the Mediterranean coast. The vegetation is composed of forest mainly calcified on crystalline soils (maquis of heath, cork-oak, maritime pine and chestnut). Direct human influence has been negligible over the past 50 years. The land use / land cover has remained almost unchanged, with the notable exception of a wildfire in 1990 that impacted a small sub-catchment. Therefore changes in the hydrological response of the

catchments are caused by changes in climate and/or physical conditions. This study investigates changes in observational data using up to 50-year daily series of precipitation and streamflow. The analysis used several climate indices describing distinct modes of variability, at inter-annual and seasonal time scales. Trends were assessed by the Mann-Kendall method. The analysis also used hydrological indices describing drought events based on daily data for a description of low flows, in particular in terms of timing and severity. The analysis

shows that there is a marked tendency towards a decrease in the water resources of the Réal Collobrier catchment in response to climate trends, with a consistent increase in drought severity and duration. But the changes are variable among the sub-catchments.

## 1 Introduction

The Mediterranean climate is known for its warm, dry summers and high precipitation events, mainly occurring

during autumn (Drobinski et al., 2014). Hence climate variability is higher in this area than in neighbouring regions. In the context of the on-going climate change, several studies have attempted to detect tendencies in the





hydrological variables. Chaouche et al. (2010) studied tendencies in the western part of the Mediterranean region over the 1970-2006 period. They mainly found an annual increase in temperature and potential evapotranspiration (PET), while no significant trends were detected for annual precipitation. Several studies

focused on the evolution of precipitation extremes over the region (Tramblay et al., 2013; Ribes et al., 2018; Pujol et al., 2007). They concluded that there had been a significant intensification in extreme precipitation events at the regional scale, while at the local scale the evolution was highly variable. Hertig and Tramblay (2017) found a widespread increase in meteorological droughts in the whole Mediterranean basin for the period 1970-2000. Lespinas et al. (2010) suggested that groundwater evolution has a significant impact on drought

trends and dynamics in the region.

In the 1960s, the French Ministry of Agriculture decided to create several experimental catchments in order to improve our knowledge of the hydrological cycle and its impact on agriculture under various French climates. One of these watershaeds is the Réal Collobrier Catchment, situated in South-East France, 50 km from the sea, at elevations ranging from 70 to 780 m a.s.l (Lavabre 1989; Folton et al. 2012). The whole watershed (area 70 km$^2$)

and several sub-watersheds have now been monitored for 50 years, including precipitation and discharge. This exceptional dataset has been recently re-checked and validated.

The objectives of this paper are to document hydrological trends over this catchment, including extreme precipitation and droughts. The study also aims to better understand drought mechanisms through precipitation and flow measurements associated with hydrological modelling. First, the physical characteristics of the

catchment, the associated dataset and the methods and model used are described. Second, annual and monthly precipitation tendencies are analysed. Finally, we discuss the 50-year evolution of the main hydrological characteristics and changes in the hydrological processes occurring in the watershed, in relation with the larger Mediterranean context.

## 2 Catchment description and data

### 2.1 General description

The Réal Collobrier research catchment is located in South-East France, at the western end of the Maures mountain range on the Mediterranean coast. Its surface is 70 km², with a maximum length of 16 km and a maximum width of 9.6 km (Fig. 1). Its elevation varies from 70 (at the outlet at the western end) to 780 m (northeast of the catchment). The average elevation is 335 m.





The Réal Collobrier watershed is representative of the geological formations of the crystalline Provence of the Maures, composed of metamorphic and granitic massifs. The main hydrographic axis is aligned with the large Collobrières East-West fault and the tributaries of the Réal Collobrier are oriented NW-SE, perpendicular to the direction of the rocks. We observe different metamorphic facies, from west to east: micaschists, amphibolites, phyllads and gneiss (Martin, 1972).

The terrains of the crystalline basement of the Maures are characterized by an alteration of the basement with a coarse and thin texture (one to two metres). Permeability is generally quite poor and groundwater supplies are extremely limited. Aquifers in this crystalline zone supply little to the Réal Collobrier tributaries that dry up rapidly in summer.

Forest cover largely dominates the hillside areas. The upstream sub-catchments therefore have a densely wooded
vegetation cover composed of most of the calcifuge Mediterranean species: cork oaks, holm oaks, heather, arbutus trees, cistus and chestnut plantations. Scrubland occupies land with a thin soil cover. Soil thickness ranges from thin skeletal ranker soil in the small Rimbaud watershed (to the east) to alluvial soil several metres thick in the low valleys. The urbanization of the catchment is low, with the exception of the small town of Collobrières (1200 inhabitants in the 1960s, 2000 inhabitants today). The downstream part of the watershed
(downstream of Collobrières) is occupied by a large agricultural plain planted mainly with vineyards. Because of the weak anthropic influence, the hydrological cycle of the watershed is very close to the natural one. The catchment is characterized by a typical Mediterranean climate with dry summers and high precipitation events, mainly during autumn (September to December). Due to the orography, the mean precipitation (1055 mm year$^{-1}$) is higher than in the surrounding areas. Table 1 summarizes the main characteristics of the catchment and sub-
catchments monitored.



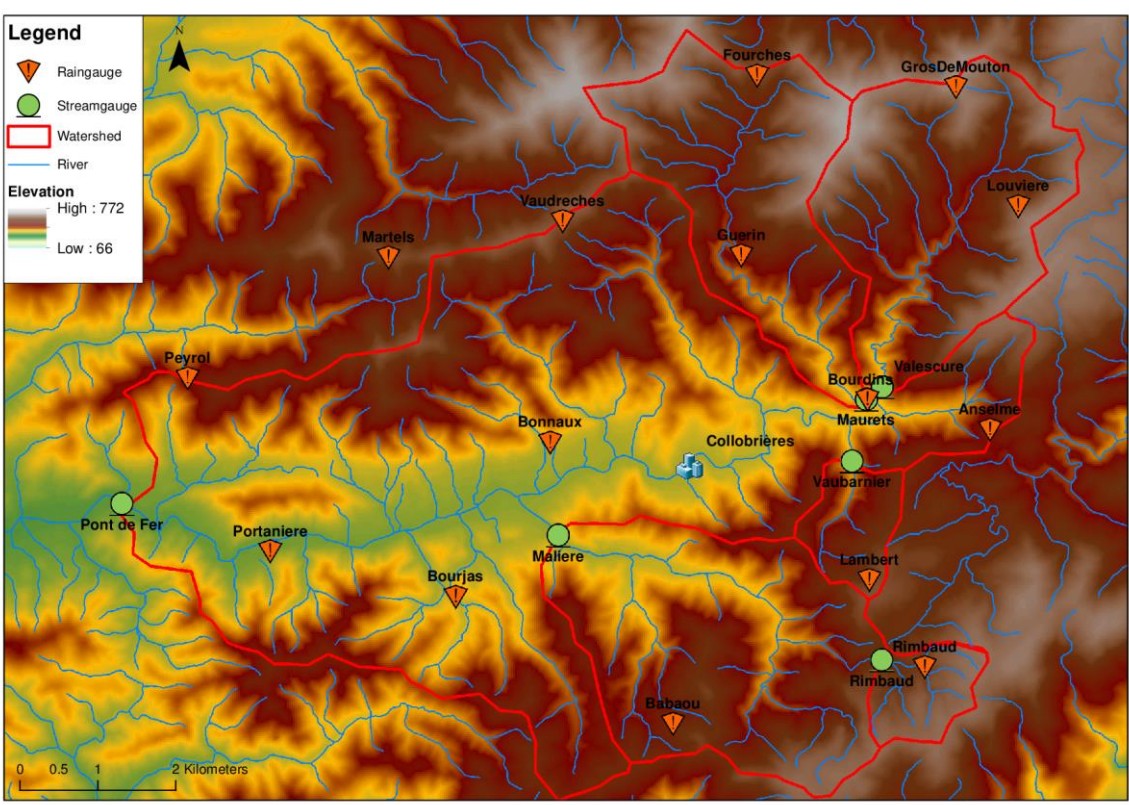

**Figure 1: Map of the whole catchment and the five sub-catchments.**

## 2.2 Meteorological and hydrological data

The analyses were conducted at the scale of the whole catchment (Outlet: Pont de Fer) and five sub-catchments. Six hydrological stations and fourteen rain gauges were used. Available data were calculated at a daily time step from January 1968 to December 2017. The equipment was slightly modified over the period analysed in order to adapt to the evolution of metrological material. Precipitations were measured by tipping-bucket rain gauges uniformly distributed over the entire catchment their spatial distribution is shown in Fig.1. The data were recorded on stable sites in terms of measurement protocols and observation conditions (approved equipment, no displacement of raingauges, and preservation of site conditions). To satisfy this condition, two raingauges were removed from the study (out of 16 measured rainfall sites, only 14 were used). The time series were subject to prior descriptive statistical reviews and contain less than 20% missing data. The trend tests were applied on 14





raingauges, but the results of the trend tests were very different from one rain gauge to another. According to
Cantat (2004), many direct and indirect rainfall factors induce a certain heterogeneity in rainfall series. He
concluded that the spatial and temporal fragmentation of the rainfall series affects the identification of long-term
rainfall trends, and that series with too many gaps make reconstruction and analysis difficult. In order to have an
overall idea of the rainfall trends on the Réal Collobrier catchment, the tests were therefore applied to the
catchment averaged rainfalls, at the risk of concealing the trend (if it exists) in the different rainfall signals.
According to Louvet et al. (2011) and given the strong density of our rain gauge network, the interpolation
method has no impact on the trend diagnostics. Spatial interpolation of the rainfall over the catchment was done
with the Inverse Distance Weighted method (IDW), widely used in rainfall interpolation (Yang et al, 2015). IDW
is based on the function of the inverse distances and assumes that each input point has a local influence that
diminishes with distance (Eckstein, 1989).

The catchment runoffs were computed from water depth measurements with hydrometric equipment (a float).
The gauging sections were stable over the period, as they were built in a calibrated structure in order to not to be
subject to change (erosion, seasonal variations in the growth of aquatic plants). The gauging sections were
equipped with triangular or rectangular sharp-crested weirs to reduce uncertainties in the river level - river flow
relation.

| Catchments | Area (km²) | Mean Elevation (m) | Number of rain gauges | Mean annual Rain (mm) | Mean Annual Runoff (mm) | Runoff coefficient | Geology | Vegetation |
|---|---|---|---|---|---|---|---|---|
| Pont de Fer | 70.4 | 335 | 14 | 993 | 271 | 0.27 | crystalline rocks with decreasing metamorphism from east to west (gneiss, micaschist, phyllads) | calcifuge vegetation (maritime pine, cork oak, chestnut, scrub) and 10% of cultivated areas (vineyard) |
| Malière | 12.4 | 386 | 3 | 999 | 334 | 0.33 | bedrock metamorphic composed of gneiss, micaschist, amphibolite and phyllad. | maquis of heath, cork-oak, maritime pine and groves of chestnut |
| Maurets | 8.5 | 453 | 3 | 1059 | 314 | 0.30 | bedrock metamorphic composed of phyllad and amphibolite | |
| Valescure | 9.2 | 466 | 2 | 1164 | 420 | 0.36 | bedrock metamorphic composed of gneiss, micaschist and amphibolite | |



| Vaubarnier | 1.49 | 391 | 1 | 1039 | 392 | 0.38 | deeply weathered schist bedrock | a well-developed forest composed mainly of chestnut trees |
|---|---|---|---|---|---|---|---|---|
| Rimbaud | 1.53 | 549 | 0 | 1099 | 629 | 0.57 | A gneiss bedrock | a dense scrubland |

**Table 1: Main characteristics of watersheds.**

### 3. Methods

This section describes the main indices used to characterize the hydrologic regime of the catchment. These indices are time series on which statistical tests were applied to determine trends.

### 3.1 Rainfall indices

We used hydroclimatic indices recommended by the ETCCDI/CRD Climate Change Indices (Peterson et al., 2001), derived from daily rainfall. These indices can be separated into four categories: threshold indices,

duration indices, absolute indices and standardized indices. Table 2 provides the detailed description of these indices. In this table "R" represents the daily rainfall and "x" is a rainfall threshold. A day is defined as "dry" (respectively wet) if the daily precipitation is below (respectively above) a threshold. For the driest days, the 2 mm threshold was used since, for water resources, small rainfalls do not reduce water stress because it is immediately taken up by vegetation (interception and evapotranspiration). Four additional thresholds were used

(5, 10, 20 and 30 mm) as proposed by Balme-Debionne (2004).

Besides analysis of the annual extremes, drought phenomena can be analysed at the monthly time step through the standardized precipitation index (SPI) (inter alia, McKee et al., 1993; Hayes et al., 2011; Lee et al., 2017; Javanmard et al., 2017). The SPI is calculated from a continuous daily precipitation time series accumulated over n sliding months (n= 1, 3, 6, 12, 24 months). These n months represent different time scales that may correspond

to periods of rainfall deficit leading to an impact on the different compartments of the water cycle. The n-month rainfall time series are fitted to a Gamma law that is usually used to fit accumulated precipitation data (Guttman, 1999; Stagge et al. 2015; Yacoub and Tayfur, 2017). This fitting is done independently for each month. Frequencies are estimated using the fitted Gamma law and transformed to a reduced and centred normal variable. This transformation yields a symmetric variable representing the n-month rainfall and called SPI. Positive SPI

values indicate precipitation above the median and negative values indicate precipitation below the median.



| Type | Name | Definition | Unit |
|---|---|---|---|
| Threshold | $WD_x$ | Annual count of wet days, when R > x (x = 2, 5, 10, 20 and 30 mm) | days |
| | $INT_x$ | Mean rainfall intensity during wet days, when R > x (x = 2, 5, 10, 20 and 30 mm) | mm day$^{-1}$ |
| | $PRTOT_x$ | Annual total precipitation from wet days when R > x (x = 2, 5, 10, 20 and 30 mm) | mm |
| Duration | $CWD_x$ | Maximum number of consecutive wet days with R > x (x = 2, 5, 10, 20 and 30 mm) | days |
| | $CDD_x$ | Maximum number of consecutive dry days with R < x (x = 2, 5, 10, 20 and 30 mm) | days |
| | $DSL_x$ | Mean number of consecutive dry days with R < x (x = 2, 5, 10, 20 and 30 mm) | days |
| Absolute | R1d | Annual Maximum one-day precipitation time series | mm |
| | R3d | Annual Maximum three-day precipitation time series | mm |
| | R5d | Annual Maximum five-day precipitation time series | mm |
| | PA | Annual total precipitation time series | mm |
| | PM | Monthly total precipitation time series | mm |
| Standardized | $SPI_n$ | Standardized Precipitation Index calculated over $n$ months ($n$ = 1, 3, 6, 12, 24) | |

**Table 2: Precipitation indices used in this study**

**3.2 Hydrological Indices**

The stationarity analysis of hydrological regimes investigated the evolution of indices calculated from daily discharges, on the six catchments of the Réal Collobrier. These indices (see Table 3) can be divided into five categories: absolute, threshold, duration, low flows and standardized indices.

The absolute indices used were the annual maximum flow, the mean annual flows and the mean monthly flows.

These indices were complemented by other indices to characterize the various aspects of hydrological drought events based on daily discharge.

The characterization of a drought episode usually depends on the definition of a threshold which characterizes the drought duration, volume deficit or similar indices (Yevjevich, 1967; Smathkin and Watkins 1997; Hisdal et al. 2004). This threshold is most often derived from the flow duration curve (FDC, Tallaksen and Van Lanen,



2004), which characterizes catchment runoff variability. The flow duration curve relates the streamflow magnitude (vertical axis) with the exceedance frequency (horizontal axis). The classical method (Beard, 1943) uses the entire series available to build a single curve. However, Vogel and Fennessey (1995a; 1995b) proposed to estimate a median flow duration curve based on yearly curves calculated from the observed series. This method is less sensitive to extreme events and is considered more representative of the mean catchment

behaviour. The choice of the threshold (a percentile of the curve) results from a compromise that allows the characterization of the low flow severity. Then, periods having values below the defined threshold are defined as periods of hydrological drought. The most commonly used percentiles range from 0.7-percentile (Q70) to 0.9-percentile (Q90) (Gregor, 2013). In the present study, we used the 0.8-percentile (Q80), a value used in many drought studies (Laaha et al., 2017).

The low flow indices were derived using this threshold. Figure 2 shows the methodology for evaluating the hydrological drought indices. These indices are:

✓ The annual volume deficit relative to the low flow threshold (called "DEF") (Giuntoli et al., 2013): it is the volume between the threshold and the flows below the threshold.

✓ The number of days per year where the flow is below the threshold (Low flow duration called "LFD"),

✓ The seasonality of low flow was studied using the concept of the centre of mass introduced by Stewart et al. (2005). The x % centre of mass is defined as the date on which x% of the annual volume deficit flows is observed. The 10 %, 50 % and 90 % centre of mass are used to define the beginning (Start), the centre (Centre) and the end (End) of the low flow period for each year.

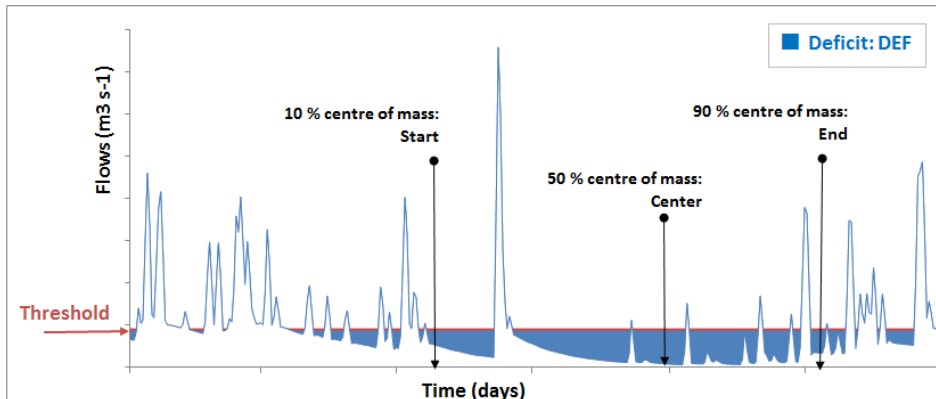

**Figure 2: Definition of hydrological drought indices**





The last index to characterize low-flow is the BFI (Base Flow Index). It was developed in the UK by the Institute of Hydrology (1980) and represents the ratio between the base flow and total discharge. We used this index to characterize the stationarity of the low-flow sensitivity of the watersheds. The BFI varies between 0 and 1. A high value of BFI reflects a high groundwater contribution to discharge, while low values are typical for

catchments with a low influence of groundwater on the total discharge. For this analysis, there are number of methods available (Brodie and Hostetler, 2005). The algorithm suggested by Lyne and Hollick (1979) has been widely used (Ladson et al., 2013) and was applied to the data.

The Standardized Streamflow Index (SSI) uses the same principle as the SPI, aggregating streamflow data over the given accumulation periods (Vincente-Serrano et al., 2012; Lorenzo-Lacruz et al., 2013). However, there is

no widely adapted probability distribution to fit streamflow data in order to calculate the SSI (Barker et al., 2016). We therefore verified the goodness of fit of the gamma distribution to empirical data for the Réal Collobrier discharge series.

| Type | Name | Definition | Unit |
|---|---|---|---|
| Absolute | AMax1d | Annual Maximum 1-day streamflow | mm |
| | QA | Annual Streamflow | mm |
| | QM | Monthly Streamflow | mm |
| Threshold | DEF | Deficit volume = Annual Streamflow below threshold | mm |
| Duration | LFD | Low flow duration = number days with Qday < threshold | days |
| | Start | Start date of drought duration | Day-of-year |
| | Centre | Centre date of drought duration | Day-of-year |
| | End | End date of drought duration | Day-of-year |
| Low-flows | BFI and BFlow | Base flow index and Base flow calculated with filter method | [-] |
| Standardized | SSI | Standardized Streamflow Index | [-] |

**Table 3: Hydrological indices used to characterize the hydrological cycle.**



### 3.3 Trend analysis methods

Trend detection tests that indicate possible non-stationarity were applied to all indices. We used two non-parametric trend tests which require only independent data, while parametric trend tests require the assumption

of independent and normally distributed data. The first order autocorrelation coefficient of the time series was calculated for each series and a test applied to detect autocorrelations significantly different from zero (function "acf" from the R package STATS, Venables and Ripley, 2002).

### 3.3.1 Mann-Kendall test and modified Mann-Kendall test

The Mann-Kendall test (MK test) (Mann, 1945; Kendall, 1975) is a rank-based nonparametric test used to detect

significant trends in climatic variables. It is widely applied on hydro-meteorological time series in different regions around the world. (Douglas et al., 2000; Yue at al., 2002; Tabari et al., 2011; Jhajharia et al., 2012; Gocic and Trajkovic, 2013; De la Casa and Ovando, 2016; Gao et al., 2017). This test, based on the ranks of observations rather than their values, is not affected by the distribution of the data and it is less sensitive to outliers. The robustness of the test was validated by several comparison tests conducted by Yue and Wang

(2004). The p-value presents the significance level to reject the null hypothesis of the Mann-Kendall test H0 (H0 = no trend). In this study, predefined significance levels $\alpha = 0.01$, $\alpha = 0.05$ and $\alpha = 0.1$ were used. The null hypothesis is rejected if the p-value is lower than the significance level chosen (Sicard, 2006). Full details related to the Mann-Kendall test can be found in the studies by Hamed (2008), Kumar et al., (2009), Gocic and Trajkovic (2013), Jhajharia et al. (2013). The function "mk.test" from the R package TREND was used (Hipel

and McLoed, 2005).

The Mann-Kendall test performs poorly when the autocorrelation is high (Yue et al., 2002; Yue and Wang, 2004; Hamed 2008). Tied values are present in samples of annual streamflow, of the standardized precipitation and of standardized flows. To compute the MK statistic in correlated series, Hamed and Rao (1998) therefore modified the Mann-Kendall test by variance correction. This modified MK test was applied to the data series for

which a significant autocorrelation was detected.

### 3.3.2 Sequential version of the Mann-Kendall Test Statistic

The sequential version of the Mann-Kendall test statistic (Sneyres, 1990) applied on time series detects approximate potential trend turning points in long term series (Kumar et al., 2016). Gerstengarbe and Werner (1999) or Bisai et al. (2014) described the successive steps to be applied in order to accept or reject the null





hypothesis (H0: Sample under investigation shows no beginning of a new trend). To apply this test, we used the

seqMK function of the PHENO RPackage (Sneyres, 1990), with a p-value of 5% to accept the H0-hypothesis.

### 3.3.3 Sen's slope estimator

The Mann-Kendall test is associated with the calculation of Sen's slope or the Theil-Sen estimator (Theil, 1950;

Sen, 1968). This method has been largely used in order to identify the slope of trend line in a hydrological time

series. Sen's slope (β) represents the median from all slopes calculated between each pair of points as follows

Eq. (1):

$$\beta = \text{Median}_{1<i<j<n} \left\{ \frac{x_j - x_i}{j - i} \right\} \qquad\qquad \text{Eq. (1)}$$

with *n* the number of data; *i, j* are indices.

The estimate of the intercept is then computed by the method recommended by Helsel and Hirsh (2002) using

Sen's slope and the median of the variables (Conover, 1980).

## 4. Hydrological characteristics of the catchment and sub catchments

### 4.1 Runoff quantitative analysis

The analysis of Table 1 and Fig. 1 allows a first comparison between catchments. The Pont de Fer watershed is

the main catchment, fed by the other sub- catchments. For the analysis we compared catchments individually or

by groups: e.g. Rimbaud and Vaubarnier catchments (two small neighbouring catchments, about 1.5 km²); the

Maurets and Valescure catchments (two neighbouring catchments having an area of around 9 km²); Malière and

Rimbaud (Malière, 12 km² receiving the flows of the Rimbaud watershed).

Figure 3 shows the interannual evolution of the mean annual rainfall and the mean annual discharge at the main

outlet of the catchment (Pont de Fer). The average precipitation is 993 mm year$^{-1}$, and ranges between 428 mm

(in 2017) and 1709 mm (in 2014). Averaged over the 50 years, the spatial variation of the sub catchments rainfall

is low (between 993 and 1164 mm year$^{-1}$, Table 1) but can be extremely high at lower timescales. The mean

annual runoff at Pont de Fer (271 mm year$^{-1}$) (Fig. 3) ranges from 35 mm (in 1989) to 805 mm (in 2011). This

main catchment receives the lowest catchment rainfall and returns also the lowest proportion of rain (runoff

coefficient of 0.27). At sub catchment scale, mean annual runoff ranges from 271 mm to 629 mm and the runoff

coefficient (runoff / rainfall ratio) is around 0.35.





The two smaller catchments have very different behaviours. The Rimbaud catchment is characterized by a runoff coefficient of 0.57, due to its special soil characteristics: the massive nature of the rocks (gneiss), the thinness of the surface formation and the morphometric characteristics promote a dynamic hydrological response to rainfall. There is a high contrast between this sub-catchment and the surrounding Vaubarnier sub-catchment (runoff

coefficient of 0.38). The Vaubarnier sub-catchment is characterized by a strong inertia due to its geological structure that plays a dominant role in the runoff production. An ensemble of juxtaposed and independent aquifers with a high retention capacity plays a strictly capacitive role, a situation which favours high losses through evapotranspiration from vegetation. This catchment can be characterized by inter annual regulations. The runoff coefficients of the other three sub-catchments (Maurets, Valescure and Malière) are equivalent, on

the order of 30 to 36 %.

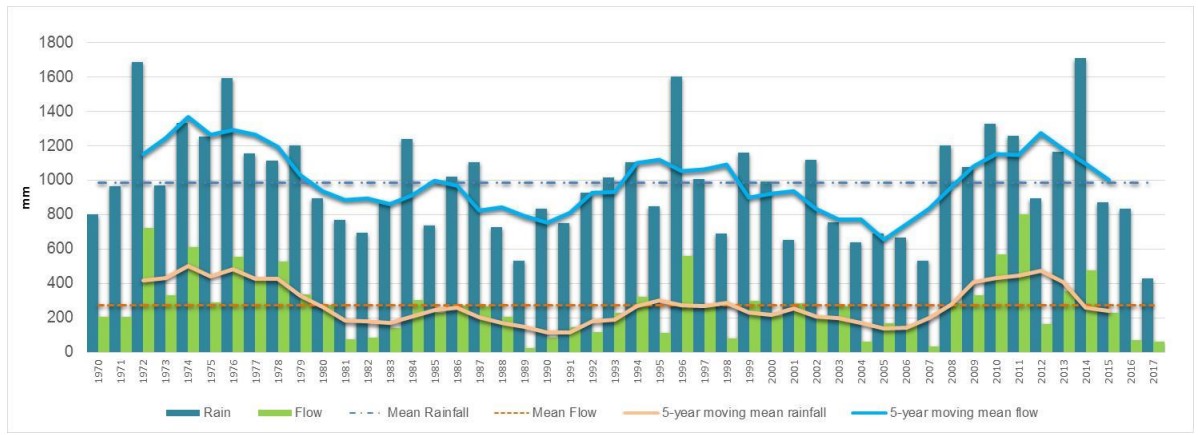

**Figure 3: Annual rainfall and flow on Pont de Fer catchment.**

**4.2 Analysis of monthly flow dynamic**

The seasonal rainfall cycle of the catchments is typical of a Mediterranean catchment (Fig. 4). In general, for all catchments, the June-August period is very dry, with rainfall of less than 40 mm month[-1] and runoff less than 10 mm month[-1]. Rainfall increases in September, but with a very limited impact on discharge. October and November are the wettest months (around 130 mm month[-1]). The wet season ranges from October to January, but high precipitation events can occur any time from September to June. The flow is not permanent and can

vanish during the dry period. At the start of the hydrological year, in September, the catchments need high rainfall contributions to reinitiate the permanent flow regime. The flows are low, in autumn, despite high rainfall. The most abundant flows then occur in winter in January with rainfall of the same order as in autumn.



If we compare the smallest catchments, the very particular hydrological behaviour of the Rimbaud sub catchment described above is clearly marked; its bare soils (gneiss) generate the highest runoff, except in

summer when it is the lowest. During this period, the Vaubarnier sub catchment has the most advantageous restitution capacity with the highest runoff, in relation with its hydrogeological characteristics already mentioned above.

If we compare the contiguous sub catchments of Maurets and Valescure, which have similar morphometric characteristics, the Maurets sub catchment is in a sheltered situation and receives slightly less rain than the

Valescure sub catchment. The excess rainfall and lithological conditions determine the differences in hydrological behaviour between the two catchments. Valescure has a higher flow linked to higher rainfall and its lithology is more suitable for flow (gneiss and mica schist). The Maurets sub catchment composed of phyllads upstream promotes a slightly more effective low water support than the Valescure sub catchment.

The Malière sub catchment of 12.3 km² includes in its upper part a lithology that is more suitable for flow

(gneiss and micachist), like the Valescure sub-catchment, but its rainfall is lower. Pont de Fer is the main catchment.

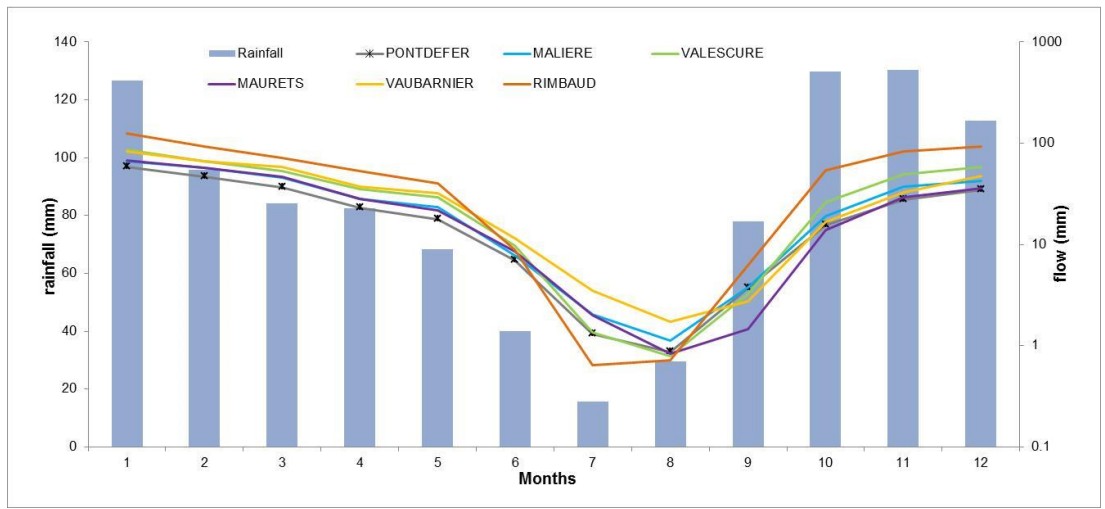

**Figure 4: Mean annual cycle of rainfall at Pont de Fer from 1968 to 2017 (histogram) and mean annual cycle of runoff for the six catchments for the same period (curves).**




**5. Tendencies**

**5.1 Rainfall trends**

**5.1.1 Absolute indices**

At the annual scale, the precipitation trend (Sen's estimator) is -5.4 mm year$^{-1}$ and is not significant according to the Mann-Kendall test. At the monthly scale, the only significant value ($P<0.1$) is March: -1.3 mm month$^{-1}$. Among the other months, eight show a decreasing trend while July, October and November indicate an increasing trend.

For annual maximum n-day precipitation, no trend was detected. This is certainly due to the difficulty of finding

tendencies on extreme values.

5.1.2 Threshold and duration indices

The indices based on daily precipitation above and below the thresholds show a drying trend over the whole 1968-2017 period (Table 4). The wet days' indices (WDx) showed a significant decreasing trend for the

thresholds ranging from 2mm to 20 mm. The results of the Sequential Mann-Kendall test statistic (not shown) showed a breakpoint in 1980. Before this date the indices were stable and afterwards a significant decrease until 2007 was detected. In the case of the Dry Spell Length (DSLx), a significant increasing trend was observed for the 5 mm and 20 mm thresholds. Before 1975, the index decreased, then a meaningful long term trend can be identified till 2007, after which it decreased again. No significant trend could be identified from the other indices

characterizing the duration or the intensity of rainfall (CWD or CDD). These indices appear to be too sensitive to sampling, since a rainstorm can disturb the calculation of consecutive dry days.

| Types | Indices | threshold 2 mm | | threshold 5 mm | | threshodl 10 mm | | threshodl 20 mm | | threshold 30 mm | |
|---|---|---|---|---|---|---|---|---|---|---|---|
| Threshold | WDx | -0.333 | ++ | -0.250 | ++ | -0.161 | + | -0.118 | ++ | -0.059 | - |
| | DDx | 0.333 | ++ | 0.250 | ++ | 0.174 | + | 0.115 | ++ | 0.054 | - |
| | INTx | -0.013 | - | -0.004 | - | 0.010 | - | 0.020 | - | 0.003 | - |
| | PRTOTx | -5.412 | - | -5.274 | - | -4.521 | - | -3.850 | - | -2.784 | - |
| Duration | CWDx | 0.000 | - | 0.000 | - | 0.000 | - | 0.000 | - | 0.000 | - |
| | CDDx | -0.029 | - | 0.000 | - | -0.094 | - | 0.167 | - | -0.050 | - |
| | DSLx | 0.032 | - | 0.068 | ++ | 0.073 | - | 0.152 | + | 0.136 | - |

**Table 4 Sen's slope of linear trends and Value of Mann-Kendall test in duration and threshold indices of rainfall. Symbols represent significance level P according to the Mann-Kendall test. (+++): P < 0.01, (++): P < 0.05, (+): P < 0.1,**
**(-): not significant.**





### 5.1.3 Standardized indices

The Mann-Kendall test and Sen's slope estimator for 1, 3, 6, 12, and 24 monthly SPI showed significant trends only for SPI-1 and SPI-3. For example, for SPI-1 the slope is -0.0005 per year, which means that SPI-1 decreased by 0.25 over the observed period. This trend can be attributed to a change in the monthly rainfall

variability related to the decrease in rainfall during March. On the other hand, the interannual variability does not seem to be significantly disturbed because the trends on SPI-6 to SPI-24 are not significant. However, for these long durations (12 and 24 months), the sequential Mann-Kendall test exhibits a continuously decreasing trend from 1980 to 2007 with a detectable change point: 1980. Over this period, the values of SPI-12 (Fig. 5) show two major dry periods centred on 1990 and during the period 2004 – 2007. In addition, the lowest values of SPI-

12, located in 1989 and 2017, are close to -3, characterising a severe dry period.

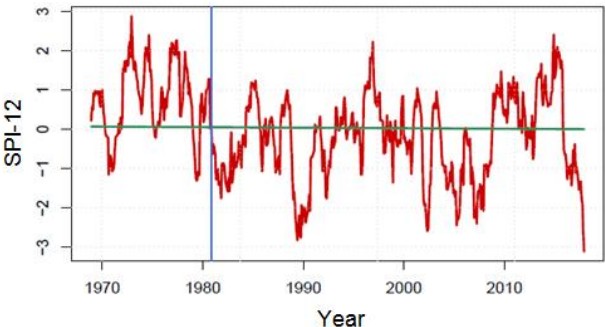

**Figure 5: SPI-12 at Pont de Fer.**

### 5.2 Hydrological trends

Yearly, seasonal and low flows of Réal Collobrier's catchments were analysed with the hydrological indices

defined in section 2.1.3.

### 5.2.1 Absolute indices

We investigated the existence of trends in mean annual, maximum annual (Amax1d) and mean monthly flows. For annual mean flows, all the detected trends were negative (ranging from -2.466 mm year$^{-1}$ at Pont de Fer to -6.357 mm year$^{-1}$ at Rimbaud) but not significant except for the Mauret catchment with a significant trend of -

4.866 mm year$^{-1}$ (Table 5). Table 5 also indicates decreasing albeit not significant, trends in maximum annual flow.




At the seasonal scale, all the catchments presented significant negative trends for spring flows, especially for March and April. Moreover, only the Vaubarnier sub-catchment presented a significant decreasing trend in September and October.

There are differences especially in the autumn months (Nov-Dec).The flow decreases in at least one of these months in the two catchments with a slow kinetic (Vaubarnier and Maurets), while in the other catchments the flows increase. However, the trends are not significant. The null values of Sen's slope observed for the summer months (July-August) are influenced by the null flows in these months, except for the Vaubarnier catchment which has the highest runoff in summer.

| | PONTFER | | MALIERE | | VALESCURE | | MAURETS | | VAUBARNIER | | RIMBAUD | |
|---|---|---|---|---|---|---|---|---|---|---|---|---|
| Jan. | -0.205 | - | -0.242 | - | -0.216 | - | -0.377 | - | -0.485 | - | -1.297 | - |
| Feb. | -0.347 | - | -0.342 | - | -0.489 | - | -0.540 | + | -0.403 | - | -1.157 | - |
| March | -0.480 | ++ | -0.522 | + | -0.570 | - | -0.872 | ++ | -0.862 | ++ | -1.304 | ++ |
| April | -0.423 | +++ | -0.368 | ++ | -0.570 | +++ | -0.561 | +++ | -0.655 | +++ | -1.080 | +++ |
| May | -0.136 | + | -0.144 | - | -0.172 | - | -0.292 | + | -0.334 | + | -0.348 | + |
| June | -0.034 | - | -0.013 | - | -0.043 | - | -0.090 | + | -0.110 | - | -0.034 | - |
| July | -0.004 | - | 0.000 | - | 0.000 | - | -0.003 | - | -0.036 | - | -0.004 | - |
| Aug. | 0.000 | - | 0.000 | - | 0.000 | - | 0.000 | - | -0.008 | - | 0.000 | - |
| Sept. | -0.002 | - | 0.000 | - | 0.000 | - | 0.000 | - | -0.040 | + | 0.000 | - |
| Oct. | -0.014 | - | -0.016 | - | 0.000 | - | -0.011 | - | -0.100 | + | 0.000 | - |
| Nov. | 0.122 | - | 0.223 | - | 0.319 | - | 0.046 | - | -0.048 | - | 0.448 | - |
| Dec. | 0.029 | - | 0.180 | - | 0.176 | - | -0.077 | - | 0.037 | - | 0.123 | - |
| Year | -1.987 | - | -3.537 | - | -2.466 | - | -4.866 | + | -3.801 | - | -6.357 | - |
| Amax1d | -0.062 | - | -0.083 | - | -0.030 | - | -0.098 | - | -0.033 | - | -0.065 | - |


**Table 5: Sen's slope of linear trends and Value of Mann-Kendall test in monthly (mm/month) and annual streamflow (mm/year) and Amax1d index. Symbols represent significance level P according to the Mann-Kendall test. (+++): P < 0.01, (++): P < 0.05, (+): P < 0.1, (-): not significant**

### 5.2.2 Low-flow and duration indices

Table 6 shows the indices computed in order to detect trends in low flows and their duration. It appears that:

- ✓ The low-flow period starts earlier for all watersheds, except the Maliere sub-catchment, and this change is significant only in the Rimbaud sub-catchment. The amplitude of this change is on average 19 days over the 50-year period observed for this catchment. This is particularly the case for the Rimbaud catchment whose functioning is strongly linked to rainfall. The decrease in spring rainfall then leads to
earlier low flows.

- ✓ The end of the low-flow period occurs later for all catchments; this shift is significant for Pont de Fer and Valescure catchments. For five of the six catchments, the dry period starts earlier and finishes later





in the year. Overall, the low flow period is longer and the low water period is centred on a later date, representing a shift of 1 to 3 weeks over the observation period.

✓   Trends on durations are significant for the two smaller catchments. The volume deficit of drought (DEF) increases for all catchments and significantly only for Vaubarnier. The Vaubarnier sub-watershed is the most strongly affected in terms of low flow deficit, followed by the Maurets sub-watershed. These are the two catchments that strongly support the low flow period (high values of the mean BFI) and are therefore more sensitive during low flow periods. Catchments such as Rimbaud,

with null flows in summer, are less impacted in terms of deficit although their period of low water increases, because of low retention capacity. The base flow (BFlow) shows a downwards trend for all catchments, and is significant for Pont de Fer and Rimbaud catchments.

   ✓   The low values of mean BFI reflect a low storage capacity (e.g. 0.21 for the Rimbaud sub-catchment) and higher values reflect a greater storage capacity (0.367 for the Vaubarnier sub-catchment). The

results are mixed, however: a decreasing trend is observed for three catchments and an increasing trend for the other three. Only one positive trend is significant (Maliere).

Overall, the results show an increase in volume deficits associated with an extension of the low-flow period. The catchments with the smallest area appear to be the most significantly affected.

|  | PONTFER |  | MALIERE |  | VALESCURE |  | MAURETS |  | VAUBARNIER |  | RIMBAUD |  |
|---|---|---|---|---|---|---|---|---|---|---|---|---|
| Start | -0.214 | - | 0.400 | - | -0.151 | - | -0.024 | - | -0.410 | - | -0.375 | +++ |
| Centre | 0.188 | - | 0.528 | +++ | 0.267 | - | 0.333 | - | 0.438 | + | 0.000 | - |
| End | 0.607 | ++ | 0.500 | - | 0.477 | + | 0.556 | - | 0.500 | - | 0.171 | - |
| LFD | 0.875 | - | 0.243 | - | 0.409 | - | 1.091 | - | 2.143 | +++ | 0.875 | + |
| DEF | 0.011 | - | 0.007 | - | 0.001 | - | 0.028 | - | 0.100 | ++ | 0.005 | - |
| BFI | -0.0003 | - | 0.0022 | +++ | -0.0002 | - | -0.0006 | - | -0.0013 | - | -0.0007 | - |
| Bflow | -0.865 | ++ | -0.873 | - | -1.162 | - | -1.591 | - | -1.748 | - | -2.339 | +++ |
| mean BFI | 0.302 |  | 0.311 |  | 0.289 |  | 0.321 |  | 0.367 |  | 0.21 |  |

**Table 6 Sen's slope of linear trends and Value of Mann-Kendall test in indices of low flows for each catchment. Symbols represent significance level P according to the Mann-Kendall test. (+++): $P < 0.01$, (++): $P < 0.05$, (+): $P < 0.1$, (-): not significant.**

### 5.2.3 Standardized indices

The results of the Mann-Kendall test and Sen's slope estimation for 1, 3, 6, 9, 12 and 24 monthly SFI over the

whole period are presented in Table 7.

The Rimbaud and Vaubarnier sub-catchments show a significant decreasing trend for all timescales, except for SFI24 for Vaubarnier. Again, the trends are significant only for the small catchments. In the other catchments,




while the trends are also decreasing, they are not significant, except for Pont de Fer for the 1 and 3 month
timescales. Significant one-month deficits (SFI-1) are doubtless related to the March rainfall deficit observed

which also translates into a deficit in March and April flows in most catchments. The sequential Mann-Kendall
test statistic detects significant change points in SFI calculated on periods above 6 months. The test indicates a
change point in 1980 for all catchments, except the Vaubarnier sub-catchment with a change point in 1989.

| SFI | PONTFER | | MALIERE | | VALESCURE | | MAURETS | | VAUBARNIER | | RIMBAUD | |
|---|---|---|---|---|---|---|---|---|---|---|---|---|
| 1 | -0.0008 | ++ | -0.0004 | - | -0.0003 | - | -0.0010 | - | -0.0012 | ++ | -0.0008 | ++ |
| 3 | -0.0009 | + | -0.0007 | - | -0.0004 | - | -0.0012 | - | -0.0013 | +++ | -0.0010 | ++ |
| 6 | -0.0009 | - | -0.0009 | - | -0.0004 | - | -0.0013 | - | -0.0014 | ++ | -0.0013 | ++ |
| 9 | -0.0009 | - | -0.0008 | - | -0.0005 | - | -0.0014 | - | -0.0015 | + | -0.0012 | ++ |
| 12 | -0.0009 | - | -0.0009 | - | -0.0006 | - | -0.0014 | - | -0.0016 | + | -0.0013 | ++ |
| 24 | -0.0010 | - | -0.0011 | - | -0.0007 | - | -0.0017 | - | -0.0019 | - | -0.0015 | + |


**Table 7: Sen's slope of linear trends and Value of Mann-Kendall test in SFI indices for each catchment. Symbols represent significance level *P* according to the Mann-Kendall test. (+++): *P < 0.01*, (++): *P < 0.05*, (+): *P < 0.1*, (-): not significant.**

**6. Discussion**

The monthly flows show a significant decreasing trend for spring flows for six catchments, in March and April. The Maurets sub-catchment is the only catchment showing a significant decrease in monthly flows for 6 months from February to June (the high flow period). This persistent decrease in these seasonal flows explains why the only significant decrease in annual flows was observed for this catchment. In addition, the Vaubarnier catchment

is the only one to present a significant decrease in September and October flows. For this catchment, the decrease in these monthly flows is confirmed by the significant increase in the low-flow volume deficit and the low-flow duration. Among the other catchments, two of them (Pont de Fer and Rimbaud) show a significant decrease in base flow (BFlow), associated with a significant lengthening of the low flow period. In this section we try to understand the reasons for this change. These trends and their variability within the whole catchment

can be explained by several factors that are detailed below.





### 6.1 Climatic factors

A decrease in flow rates is often related to a decrease in rainfall. However, annual rainfall has not exhibited any trend, as reported in Chaouche et al. (2010), who did not detect any significant trends in annual precipitation in the Mediterranean region during 1970-2006. Only a significant downward trend in March precipitation, with a

decrease in wet days (thresholds between 2 and 20 mm day$^{-1}$) and a longer dry spell was detected in the catchment. These factors can have a direct impact on the decrease in flows and changes in the genesis of runoff. Evapotranspiration is a key flux controlling the surface water balance. It is basically determined by the potential evapotranspiration (PET), which represents the climatic demand, and the upper soil wetness, which represents the actual amount of water that can be mobilized. In order to evaluate the climatic demand, temperatures from

the nearest Météo-France climatological station in the reference homogenized French climatic network (Le Cannet des Maures) from 1959 to 2011 and the monthly PET calculated by the Penman-Monteith (Allen et al., 1998) formula for the Toulon station from 1959 to 2014 were studied (Table 8).

| | PET (mm year$^{-1}$) | | Tmin (°C year$^{-1}$) | | Tmoy (°C year$^{-1}$) | | Tmax (°C year$^{-1}$) | |
|---|---|---|---|---|---|---|---|---|
| Jan. | 0.069 | - | 0.007 | - | 0.013 | - | 0.020 | - |
| Feb. | 0.105 | ++ | -0.013 | - | 0.003 | - | 0.024 | - |
| March | 0.171 | ++ | 0.008 | - | 0.026 | ++ | 0.036 | +++ |
| April | 0.098 | - | 0.020 | ++ | 0.024 | + | 0.023 | ++ |
| May | 0.133 | - | 0.043 | +++ | 0.043 | ++ | 0.047 | +++ |
| June | 0.238 | +++ | 0.038 | +++ | 0.043 | +++ | 0.049 | +++ |
| July | 0.250 | +++ | 0.044 | +++ | 0.037 | +++ | 0.027 | ++ |
| Aug. | 0.100 | - | 0.050 | +++ | 0.048 | +++ | 0.054 | +++ |
| Sept. | 0.232 | +++ | 0.016 | - | 0.019 | + | 0.029 | + |
| Oct. | 0.100 | + | 0.035 | ++ | 0.027 | ++ | 0.017 | - |
| Nov. | 0.086 | + | 0.017 | - | 0.019 | - | 0.018 | + |
| Dec. | 0.047 | - | 0.031 | + | 0.022 | + | 0.013 | - |
| Year | 1.576 | ++ | 0.007 | - | 0.031 | +++ | 0.039 | +++ |

**Table 8: Sen's slope of linear trends and Value of Mann-Kendall test in PET at Toulon and T at Le Cannet des**
**Maures. Symbols represent significance level P according to the Mann-Kendall test. (+++): P < 0.01, (++): P < 0.05,**
**(+): P < 0.1, (-): not significant.**

At the annual scale, the mean temperature shows significant changes with +1.6°C from 1959 to 2011, i.e. +0.3°C per decade. This value is consistent with those of Chaouche et al. (2010) who noted the same rates in the French Mediterranean region and with Lespinas et al. (2010) who observed an increase of +1.4°C from 1965 to 2004 in

Languedoc-Roussillon. At the monthly scale, there is a significant increase in temperature in the spring and





summer months. The months with the strongest warming are August and May. The temperature also increased in winter (December) but with a lower magnitude. Annual potential evapotranspiration (PET) displays an upward significant trend with an amplitude of +1.6 mm year$^{-1}$, confirmed by the results of Chaouche et al. (2010) who found a trend varying from +1 to +4 mm year$^{-1}$. Significant changes in monthly PET show upward trends in

some months. No trends are seen in monthly PET in April, May, or August whereas significant trends in monthly temperature are observed. These discrepancies can be explained by differences in the local climatic conditions, Toulon being at the seashore, south of the catchment, while the Cannet des Maures is situated inland, north of the catchment.

To combine PET and soil wetness, the GR4J model (Perrin et al., 2003) was applied over the whole catchment

(Pont de Fer) from 1968 to 2015. Tendency tests were applied to the annual time series of the mean reservoir levels that represent the amount of water in the upper soil and showed a decrease, with a particularly low level in 1989 and 2007. In addition, outputs from a reanalysis of soil wetness by the Safran-Isba-Modcou model (Habets et al.; Vidal et al., 2010) were tested, showing the same significant decrease in soil wetness. Hence, it is clear that the increase in evaporation demand, combined with a rainfall decrease in March tend to dry the upper soil,

reducing the water supply to the different catchments.

### 6.2 Catchment characteristics and functioning

There are common trends between the catchments, such as a greater severity of the low flow period, but the response differs according to the sub-catchment.

Firstly, the size of the watershed seems to be an explanatory factor for these differences. For example, the smaller the catchment the more sensitive it is to climate variation. This is the case of the Rimbaud and Vaubarnier catchments, although their sensitivity to climate variation is totally different. Secondly, the physiographic characteristics of the catchments (topography, geology, lithology, etc.) imply different sensitivities of the catchments to climatic changes. Thirdly, the water storage capacity of watersheds seems to be

an important explanatory factor of the different impacts of climatic variations on flows. The greater the storage capacity of the catchment or the slower the dynamics of the catchment, the less the low flows will be affected by a decrease in rainfall and an increase in temperature. This is explained by the decrease in underground flows and their contribution to base flows. This is the case of the two catchments of Maurets and Vaubarnier, which have the lowest dynamics and are characterized by a higher infiltration capacity.





## 6.3 Land use

Trends in flows may also be due to changes in land use. As many studies have shown, the evolution of land use over the years can affect the regime of rivers (Hibbert 1967; Hewlett 1982; Andréassian 2004; Brown et al. 2005; Ssegane et al. 2013). These disturbances are generally caused by human actions, such as urbanization, deforestation, agriculture, or by natural causes such as fires. This point was disregarded in our explanations because the Réal Collobrier watershed is a forest catchment area that has been weakly impacted by human activity for the last 50 years. The only urban area (the village of Collobrières) has not extended much and is located downstream of the six sub-catchments studied. Only the Rimbaud catchment was burnt by forest fires in 1990 (80 % of its surface). Studies on the impact of this fire on the flows showed that only the flood dynamics were impacted during a period of four years after the fire (Folton et al., 2015; Lavabre et al., 1998, 2000). No impact was observed on the annual flows or low flows in this catchment. These arguments suggest that the physiographic and land cover characteristics of the watershed have remained relatively stable over the last 50 years and cannot be considered an explanatory factor for the flow trends.

## 7. Conclusion

The analysis of the 1968-2017 period of the Réal Collobrier catchment data shows a decrease in rainfall in March and a decrease in rainfall duration.

With the observed increase in temperatures and PET, climatic factors show a tendency to decrease the amount of water available for watersheds. This is observed globally in all of the sub-watersheds studied, especially in spring and during the low flow period.

Hydrological factors modulate the climatic trends among the sub-catchments. Depending on their hydrological functioning, the impact of a climatic change leads to responses that can be highly contrasted. This study shows that the catchments that usually support low flows are the most sensitive to climatic disturbances.

The fine-scale hydrological information obtained thanks to the Réal Collobrier research catchment has enabled us to observe that at this fine scale the impact of a climatic variation on the hydrology is much more variable than the variability imposed by the climate. The rainfall-runoff relationship is complex and multiple nonlinearities (between flow and climatic variables such as rainfall or temperature), make it difficult to understand and to quantify the influence of the detected change in climate on the water resource. However, this appears to be specific to small watersheds. This has to be considered as an important result too as it underlines the complex functioning at small scale. The response of a particular watershed to climatic change integrates not

only the climate input, but also changes in storage, as well as change and variability in the hydrology of the
watershed.

The reliable evaluation of trends requires the availability of data in catchments that are near-natural and unregulated, contain long records lengths, are active gauged catchments and have good-quality data. The Réal Collobrier research catchment is part of the French reference hydrological observatory (http://www.ozcar-ri.org/real-collobrier/) and is particularly suitable for this assessment. These catchments constitute an
exceptionally valuable data source to effectively identify, quantify and interpret hydrological change in an area weakly influenced by human activities. Most studies in the literature analysing flow variability (e. g. Giuntoli et al. 2013; Lespinas et al. 2010), are based on samples of catchments with few small catchments (less than 10 km²). Hence the data on the research catchment of Réal Collobrier allow us to study the flow variations in these small catchments in connection with the climate.

The measures in the catchments will be continued and enriched in the future according to need. These measures will be completed by modelling to better document climate-related drivers of change. Concerning hydrological functioning and land use changes, reforestation experimentation on the Malière catchment planned in the future will be analysed.

**Competing interests**

Authors declare that no competing interests are present.

**Author contribution**

Tolsa M. collected the data.  Folton N. and L'hermite P. conceived and designed the analysis. Folton N. performed the calculations and analysed the results. Arnaud P. and Martin E. contributed to the interpretation of the results. Folton N., Martin E., Arnaud P. wrote and prepared the manuscript with contribution from all co-
authors.

**Acknowledgements**

Data from the Safran-Isba-Modcou was provided by François Besson (Météo-France, Direction de la climatologie et des services climatiques). Data from the Cannet des Maures Station (temperature) and Toulon (PET) are provided by Météo-France.



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
