# Peer review of "A 50-year analysis of hydrological trends and processes in a Mediterranean catchment"

_Hydrology and Earth System Sciences, 2018_

## Referee Comment (RC1) · Anonymous Referee #1 · 6 Dec 2018

**1 General comments**

This article presents an analysis of hydrometeorological trends in the Réal Collobrier observatory, which gathered 50 year long time series of precipitation and streamflow.

The catchment is small, 70 km2, but it is well observed and it is scarcely influenced by human activities, thus, the observed changes are due to climate and environmental changes, instead of human activities.

The authors calculate well-known indices (extremes and drought) and then use the Man Kendall method in order to calculate trends.

In order to explore the causes of these trends, they also analyze trends in PET (calcu-

lated from observations) and modeled soil moisture (with two different models).

They find that water resources are diminishing in this study area. The most significant trends are in spring. As there are no significant trends of precipitation, the main cause is the increase in PET. They also find that it is difficult to detect significant trends (except in spring). These results are coherent with what is in general seen in other Mediterranean basins.

They also find that there is high variability between subbassins. This result is important for the community that studies the impacts of climate change on water resources, as it shows that the impacts of climate change depend on the physical characteristics of the basins, whose spatial variability is high at fine scales and, thus, shows that it is difficult to study the impacts of climate change at the scales of human activities (1-10 km).

This article also shows the importance of having well instrumented long-term hydrometeorological observatories, something that is necessary, but rarely done, as long-term funding for observational activities is scarce, specially in the Mediterranean areas, where research funds are scarce in general.

The article is well written, well organized, and its methodology is sound. Its results are relevant and an original contribution to its field of research. Therefore, I recommend its publication with minor changes.

**2 Specific comments**

1. English is not my native language, but I suspect that "tendencies" is not the right word to mean "trends".

2. Line 69: There are vineyards in the basin. Being France, I guess they are not irrigated. As nowadays vineyards are being equipped with irrigation in other Mediterranean areas, to irrigate during droughts, it would help the reader to mention if

these are, or not, irrigated.

3. Line 126: I wonder if the authors checked the goodness of fit of the gamma law. Later, they did for standardized streamflows, so I'd like to see a comment on this fact.
* * *

---

## Referee Comment (RC2) · Anonymous Referee #2 · 4 Feb 2019

This paper represents a good example of a comprehensive trend analysis based on a highly gauged mediterranean basins. The analytical tools used for the analysis are widely recognized standards and results reflect observations made in many other sites. Nevertheless the investigation is, in my opinion, worth of publication. I have only a few suggestions for the authors in order to provide a more clear presentation.

1. More details could be given with regards to procedures used to evaluate the BFI. The authors only mention related papers but, since there is a large number of methods and techniques for filtering the streamflow data, I believe that this topic may deserve a sub-section.

2. More details could given also to the interpretation of trend analyses on SPI and SSI. Being these basically two standardized quantities, their trend should involve changes

in the second order statistics of variables which are seldom investigated.

3. The conclusion section is too loose. The authors should here reinforce their statements and beliefs on cause-effect dynamics going on at the hydrological level.

---

## Author Comment (AC3) · 1 Mar 2019

Dear Giuseppe Tito Aronica,

We would like to thank the two anonymous reviewers for the positive feedback and the advice of minor revision to our manuscript. The comments are encouraging. As suggested, we have modified some sections. We hope you will find our revised manuscript suitable for publication.

Yours sincerely,

Nathalie Folton on behalf of the authors

Please also note the supplement to this comment:

[Figure]

https://www.hydrol-earth-syst-sci-discuss.net/hess-2018-547/hess-2018-547-AC3-supplement.pdf

[Figure]

**Supplement:**

Dear Giuseppe Tito Aronica,

We would like to thank the two anonymous reviewers for the positive feedback and the advice of minor revision to our manuscript. The comments are encouraging. As suggested, we have modified some sections. We hope you will find our revised manuscript suitable for publication.

Yours sincerely,

Nathalie Folton on behalf of the authors
* * *

[revised manuscript text omitted]

---

## Author Response (AR1)

**Reviewer #1**

We would like to thank the reviewer #1 for careful and thorough reading of the manuscript. We appreciate his positive feedback. And as suggested by the reviewer, we will add the information about irrigation and the goodness of fit of the gamma law for SPI. The suggested correction for the word "trends" will be made.

**Reviewer #2**

We would like to thank reviewer #2 for his positive feedback on our paper and these constructive suggestions. See below the detailed responses to the comments sent.

1. More details could be given with regards to procedures used to evaluate the BFI. The authors only mention related papers but, since there is a large number of methods and techniques for filtering the streamflow data, I believe that this topic may deserve a sub-section.

> We understand the point of the reviewer. With regards to the procedures used to evaluate the BFI, and as suggested by the reviewer, we will improve section 3.2. A sub-section 3.2.3 "low flow indices" will be added to discuss BFI calculations. We have considered two different approaches: the first method is the basic hydrograph separation proposed by the Institute of hydrology (1980) (already mentioned in the manuscript) and the second approach uses a recursive digital filter suggested by Lyne and Hollick (1979) to separate the hydrological processes. The results of these different approaches have been analysed on the catchments. The trends on mean BFI are very similar and regardless of the calculation method. These additional details will therefore be included in the new subsection 3.2.3.

> 2. More details could given also to the interpretation of trend analyses on SPI and SSI. Being these basically two standardized quantities, their trend should involve changes in the second order statistics of variables which are seldom investigated.

Concerning the remark on the analysis of trends on SPI and SSI, it is true that second-order statistics of variables are not included in the analysis because of the sampling problems. In order to give more details about the interpretation of trend analyses on SPI and SFI, a homoscedasticity test was realised by the Bartlett's test (1937). This test is used to test if different samples have equal variances. We verified the homogeneity of variances for SPI and SFI accumulated on 1 month to 24 months (with a p-value of 5% to accept H0 hypothesis: Identical variance). We compared the variance calculated over the first 25 years (P1) and over the last 25 years (P2). The SPI variance was found homogeneous over the two periods with the exception of the SPI24, for which the P1 variance is higher than P2 variance. As for the previous results on the SFI, the results the homoscedasticity test on SFI depend essentially on the hydrological functioning. So, the catchments characterized by a hydrological functioning mainly controlled by precipitations (Rimbaud, Valescure) show a significant increasing of SFI variability, for SFI accumulated on 1 month to six months. This suggests that these watersheds are more influenced by decreasing precipitation, creating more of a gap between high and low water periods. For the catchments characterized by a lower dynamic (Vaubarnier and Mauret), the test indicates a significant decreasing of SFI variability for longer accumulation (SFI12 and SFI24). This is

probably related to the presence of longer periods of drought over the period P2, reducing the variability of mean flows.

These additional results provide input to the analysis and are added in the text in the discussion section to improve the manuscript.

3. The conclusion section is too loose. The authors should here reinforce their statements and beliefs on cause-effect dynamics going on at the hydrological level.

We agree with reviewer #2 that the conclusion section is somewhat vague and we have revisited the text. Additional elements on the different response of the various sub watersheds to climate change will be added. Hence the importance of the hydro-geo-logical factors will be highlighted to explain the variability of observed changes in the catchment.

Dear Giuseppe Tito Aronica,

We would like to thank the two anonymous reviewers for the positive feedback and the advice of minor revision to our manuscript. The comments are encouraging. As suggested, we have modified some sections. We hope you will find our revised manuscript suitable for publication.

Yours sincerely,

Nathalie Folton on behalf of the authors

[revised manuscript text omitted]